# Leg Dominance Effects on Postural Control When Performing Challenging Balance Exercises

**DOI:** 10.3390/brainsci10030128

**Published:** 2020-02-25

**Authors:** Arunee Promsri, Thomas Haid, Inge Werner, Peter Federolf

**Affiliations:** 1Department of Sport Science, University of Innsbruck, Innsbruck 6020, Austria; arunee.pr@up.ac.th (A.P.); thomas.haid@gmx.net (T.H.); inge.werner@uibk.ac.at (I.W.); 2Department of Physical Therapy, University of Phayao, Phayao 56000, Thailand

**Keywords:** leg dominance, one-leg stance, motor control, movement strategy, diagonal movement, balance board, sex difference, minimal intervention principle, optimal feedback control theory, principal movement, principal component analysis (PCA)

## Abstract

Leg dominance reflects the preferential use of one leg over another and is typically attributed to asymmetries in the neural circuitry. Detecting leg dominance effects on motor behavior, particularly during balancing exercises, has proven difficult. The current study applied a principal component analysis (PCA) on kinematic data, to assess bilateral asymmetry on the coordinative structure (hypothesis H1) or on the control characteristics of specific movement components (hypothesis H2). Marker-based motion tracking was performed on 26 healthy adults (aged 25.3 ± 4.1 years), who stood unipedally on a multiaxial unstable board, in a randomized order, on their dominant and non-dominant leg. Leg dominance was defined as the kicking leg. PCA was performed to determine patterns of correlated segment movements (“principal movements” PM_k_s). The control of each PM_k_ was characterized by assessing its acceleration (second-time derivative). Results were inconclusive regarding a leg-dominance effect on the coordinative structure of balancing movements (H1 inconclusive); however, different control (*p* = 0.005) was observed in PM_3_, representing a diagonal plane movement component (H2 was supported). These findings supported that leg dominance effects should be considered when assessing or training lower-limb neuromuscular control and suggest that specific attention should be given to diagonal plane movements.

## 1. Introduction

In humans, the preference of using one side of the body for unimanual tasks is well recognized as a self-evident aspect of human motor control. This phenomenon reflects bilateral asymmetry in motor control circuitry between the right and left hemispheres, seen in both upper and lower extremities [1,2]. For the upper limb, handedness is typically clearly defined by the ability to write competently. The development of handedness is often attributed to distinct specializations of the underlying neural control system. For example, the dynamic dominance hypothesis by Sainburg purports that the dominant system specializes in controlling limb trajectory or predictive control, while the non-dominant specializes for control of limb position or impedance control [1,3,4]. However, there are also recent results challenging the idea of interlimb differences in control strategies [5,6]. 

Unlike the upper limbs, the lower limbs have the inherent functions of supporting the body weight and performing locomotion, which requires effective coordination between legs. Possibly as a result, footedness seems to be much more task-dependent than handedness. Several studies reported that the preferred leg differs between dynamic tasks, e.g., kicking a ball, and static tasks, e.g., balancing [7,8,9,10,11]. Footedness is also interesting from an epidemiological point of view, since leg dominance has been reported as an internal risk factor for sports-related lower-limb injury [12,13,14,15].

Traditionally, studies on footedness analyzed unipedal postural stability by assessing the center of pressure (COP) oscillations; however, the COP-based variables, such as sway area [16], COP excursion [17], 95% COP elliptical area [18], standard deviation [18], total path length [16,18], and speed [8,18], did not indicate an effect of leg dominance. Only in a body-tracking test [19] were effects of footedness observed when participants received visual feedback of their performance. Visual feedback might make a difference, since processing visual information is also distributed asymmetrically between the brain hemispheres [20,21], which could interact with neural asymmetries, causing motor control laterality. Hence, it seems that, without receiving external feedback, analyzing the COP position might be insufficiently sensitive for detecting bilateral asymmetries in postural control. One possible reason why conventional methods often fail to detect the influence of leg dominance on postural control is that the COP motion is a resultant variable that condenses information on the whole-body posture and information on postural accelerations into one two-dimensional (2D) variable [22]. The COP motion alone does not reveal the actual coordinative structure or the control of the segmental movements facilitating postural control [23,24]. This is a critical flaw, since recent advancements in our understanding of motor control [25,26,27] suggest that in higher-dimensional movements (“higher dimensional” in the sense of many degrees of freedom available for the motion [28]), the control system distinguishes between movement dimensions that are task-relevant and dimensions that are not task-relevant. This well-established concept for motor coordination is known as the *minimal intervention principle* (MIP) and can be seen as an outcome of the *optimal feedback control theory* [25]. Specifically, the MIP suggests that the neuromuscular system tightly controls movement variations/dimensions if they affect the task goal; otherwise, movement variations are largely ignored, since making corrections is expensive in terms of control-dependent noise and energy costs [29,30]. In other words, different control behavior can be expected [11,31] in different movement dimensions. However, since the COP condenses the multidimensional information on postural control into only two dimensions [22], it is far less sensitive for such underlying mechanisms. In fact, improved sensitivity for differing motor control characteristics through consideration of the MIP was already documented in the context of aging effects [32], effects of concurrent dual tasking [31,33], or effects of sensory perturbations [34] affecting postural control.

In two recent studies, we also investigated the effects of leg dominance on postural control by decomposing unipedal balancing movements into sets of movement dimensions through a principal component analysis (PCA). As predicted by the MIP, interlimb differences in the control of movement components were observed in specific movement components, and particularly in such components that were critical for the given task of maintaining stability [11,35]. In the second study, however, we found significant effects only in a stable condition, but not when standing on an unstable foam [35]. Since training on unstable support surfaces is an important therapeutic exercise, with proven effectiveness in facilitating and regaining neuromuscular control [36,37,38,39,40], investigating unipedal balancing movements on a multiaxial unstable platform is of particular interest and may provide clinically relevant information for injury prevention and rehabilitation. 

In summary, the current study used a PCA approach to investigate the effects of leg dominance when unipedally standing on a multiaxial unstable board. Specifically, we hypothesized that bilateral asymmetry might emerge as a difference in the coordinative structure (hypothesis H1), detectable as a difference in the composition of the postural movement components (PC-components) that produce, in combination, the whole-body balancing movements. Second, bilateral asymmetry might emerge as a difference in how individual movement components are controlled (hypothesis H2). Additionally, we also tested for sex effects and interactions between sexes and leg dominance.

## 2. Materials and Methods

### 2.1. Participants

Twenty-six physically active young adults who reported no neuromuscular problems and who did not participate in balance-specific training within the last six months volunteered for the current study. Their characteristics are summarized in Table 1. Leg dominance was defined as the preferred leg for kicking a ball, since this definition had proven most effective in determining interlimb differences in unipedal postural control [11,35]. Most participants (96%) self-reported right-leg dominance. The leg dominance of all participants coincided with their hand dominance. All volunteers provided informed written consent prior to their participation, and the study protocol was approved by the Board for Ethical Questions in Science of the University of Innsbruck, Austria (Approval Code No.: 14/2016). 

### 2.2. Experimental Equipment

An 8-camera motion-tracking system with a sampling rate of 250 Hz (Vicon Bonita B10 with Nexus 2.2.3 software; Vicon Motion Systems Ltd., Oxford, UK) recorded the trajectories of 39 reflective markers. The markers were attached to the participant’s skin by hypoallergenic double-sided tape, and they were distributed over the volunteers’ bodies according to the “Plug-In Gait” marker setup (Vicon Motion Systems Ltd., Oxford. UK). The magnitude of error of the Vicon system in the dynamic tasks is less than 2 mm [41]. A multiaxial unstable platform was used in this study (MFT Challenge Disc; Trend Sport Trading GmbH., Austria) that consisted of an upper 40 cm diameter round plate connected to a circle base plate by a group of four rubber cylinders at the center of the board, allowing it to tilt in all directions (stance height = 8 cm; weight = 3.6 kg). To standardize the position of the board, the center of the board was placed over the center of a reticle cross-line marked on the floor, to which the two main axes of the board were perpendicularly aligned. Hence, the same fulcrum position was furnished for all balancing trials of all subjects. 

### 2.3. Experimental Procedures

Participants started with a 15-second familiarization trial on the balance board, with no instruction and no feedback. Then, on each foot, volunteers performed one 80-second balancing trial, barefoot, on the balance board. The order of trials (right foot–left foot/left foot–right foot) was randomized. All participants were asked to stand on one foot, with hands akimbo, by placing the stance foot (base of the 2nd metatarsal bone) over the center of the board and keeping the second toe on an anteroposterior midline of the board. The hip and knee of the lifted leg were flexed at 20 and 45 degrees, respectively [11]. The volunteers focused their gaze on a target (10 cm in diameter red circle) placed at the individual participant’s eye level, on the wall, approximately five meters away. During testing, participants were instructed to try to keep the board horizontal, to not touch the stance leg with the lifted leg, to avoid any movements not required for balancing such as scratching, and to look straight ahead and keep their eyes on the target. After the first trial, participants could rest (seated) at their own discretion, for up to three minutes, before the second trial started.

### 2.4. Data Analysis

#### 2.4.1. Kinematic Data Pre-processing

All data processing was conducted in Matlab (MathWorks Inc., Natrick, MA, USA). The preprocessing steps and the PCA analysis were conducted based on earlier studies [11,22,35,40]. Any gaps in marker trajectories were filled by a PCA-based reconstruction technique [42,43]. One PCA analysis was conducted after normalizing, weighing, and pooling the middle 60 s of each balancing trial. The middle 60 s were selected to avoid movements relating to settling in or impatience at the end of the balance task. The following preprocessing steps were conducted: nine asymmetrical markers placed on upper arms, lower arms, right scapular, upper thighs, and the lower thighs were omitted [11,35,40]. If participants stood on the left-leg, then their maker data were mirrored and relabeled such that all input data for the PCA appeared as right-leg trials; thus, all datasets could be concatenated and then only one PCA could be computed. Projecting all data onto the same set of eigenvectors is a prerequisite for direct comparisons between trials to identify differences in coordination or control between the dominant and non-dominant legs [11,35]. The 3D coordinates (x, y, z) of the remaining 30 markers of each dataset at a given time, *t*, were interpreted as 90-dimensional posture vectors [22,44,45,46]:(1)$$\vec{p\left(t\right)}\;=\;[x_{1}\left(t\right),\;y_{1}\left(t\right),\;z_{1}\left(t\right),\ldots ,x_{30}\left(t\right),\;y_{30}\left(t\right),\;z_{30}\left(t\right)],$$

Then, all marker data were centered by subtracting the mean posture vector [44,45,47], thus preventing differences in mean marker positioning in space from having an influence on the PCA outcome [22]. For each subject, *subj*, a mean posture vector was calculated:(2)$$\vec{\bar{p^{subj}}}\;=\;[\bar{x_{1}},\;\bar{y_{1}},\ldots ,\bar{z_{30}}],$$
where the bar over the variable indicates the mean over time, $\bar{x}=mean_{t}\left(x\left(t\right)\right)$, and was subtracted from each posture vector: (3)$$\vec{p^{\prime }\left(t\right)}\;=\;\vec{p\left(t\right)}\;-\;\vec{\;\bar{p^{subj}}}.$$

Thus, the PCA was conducted on deviations from a subject’s mean posture, i.e., on postural movements, and not on the postures themselves. This procedure is a first step toward removing anthropometric differences [22]. 

The centered posture vectors were normalized by using the mean Euclidean distance, $\bar{d^{subj},}$ to address anthropometric differences [22], i.e., for each subject, *subj*, the postural movement vectors $\vec{p^{\prime }\left(t\right)}$ were divided by their Euclidian norm, $d^{subj}\left(t\right)=\sqrt{x_{1}{\left(t\right)}^{2}+y_{1}{\left(t\right)}^{2}+z_{1}{\left(t\right)}^{2}+\ldots +z_{30}{\left(t\right)}^{2}}$**,** averaged over time:(4)$$\vec{p^{\prime \prime }\left(t\right)}\;=\;\;\frac{1}{\bar{d^{subj}}}\;\vec{p^{\prime }\left(t\right)},$$

The normalized posture vectors were weighted by considering sex-specific mass distributions for yielding comparison between the sexes [48,49]. Specifically, for each marker, *i*, a weight factor, $w_{i},$ was calculated by dividing the relative weight of the segment [48] to which the marker was attached, *m*_s_, by the number, $n_{s}$, of markers on this segment [49]. For markers placed on joints, the masses of both segments were added. For example, $w_{i}$ for the knee markers was calculated as $w_{i\;}=m_{thigh}/n_{thigh}+\;m_{shin}/n_{shin},$ with $n_{thigh}=\;n_{shin}=3,\;\;m_{thigh}=14.16\%$, and $m_{shin}=4.33\%$ for men [48]. Thus, the normalized postural movement vectors had the following form:(5)$$\vec{\;p^{\prime \prime \prime }\left(t\right)}\;=\;\frac{1}{\bar{d^{subj}}}\left[w_{1}\left(x_{1}\left(t\right)-\;\bar{{x_{1}}^{subj}}\;\right),\;w_{1}\left(y_{1}\left(t\right)-\;\bar{{y_{1}}^{subj}}\;\right),\ldots ,w_{30}\left(z_{30}\left(t\right)-\;\bar{{z_{30}}^{subj}}\;\right)\right],$$

Then, the normalized posture vectors, $\vec{p^{\prime \prime \prime }\left(t\right)},$ of all balancing trials of all participants were concatenated to form one common input matrix, consisting of 780,000 posture vectors (26 subjects * 2 trials * 15,000 frames) of 90 dimensions (30 3D-marker coordinates) for the PCA. 

#### 2.4.2. Principal Component Analysis

The PCA was computed through a singular value decomposition of the input matrix’s covariance matrix [44,45,47]. The PCA yielded a set of eigenvectors, $\vec{PC_{k}}$, which form a new basis for the vector space of marker positions, i.e., the eigenvectors provide a basis in which changes in posture (“postural movements”) can be quantified [44,45,47]. Thus, we replace the intuition-driven definitions of movement strategies described in many previous research studies, e.g., “ankle strategy, hip strategy” [50,51,52], through a data-driven vector formalism [22,24]. Since the PCA eigenvectors represent correlated changes in the underlying data, the eigenvectors can be interpreted as patterns of correlated segment movements, as “movement strategies” [22,24] or “principal movements” (PMs) [23]. Through projecting the normalized posture vectors recorded for each trial onto the eigenvector basis, we obtained the PCA scores, which we call “principal positions” PP_k_(t), where t denotes the time and k denotes the order of the PM: (6)$$PP_{k}\left(t\right)\;=\;\vec{p^{\prime \prime \prime }\left(t\right)}\;\cdot \;\vec{PC_{k}},$$

The PP_k_(t) quantify how much the posture at time, t, deviates from the mean posture, according to the pattern defined by the k-th eigenvector. Thus, the PP_k_(t) represent the amplitude of each movement component, PM_k_. The term “principal” in the variable name indicates that the variable was derived from the PCA, i.e., that the variable is defined through the PCA eigenvector basis. 

In analogy with Newton’s mechanics, the PP_k_(t) can be differentiated to obtain “principal velocities” and “principal accelerations”, PA_k_(t) [22]. The PA_k_(t) quantify the acceleration of the postural movements and thus represent a variable that quantifies how the mechanical system is controlled [23]. It can be mathematically proven that the acceleration of the original marker trajectories can be derived from the sum of all PA_k_(t) [53]; thus, the PA_k_(t) are physically consistent variables, representing the accelerations of the body segments. In unperturbed balancing trials, there are only two types of forces that produce postural accelerations: gravity (which is constant) and forces produced by the sensorimotor system. In other words, together, the PA_k_(t) represent the resultant control variables through which the postural control system changes its kinematic state [40]. In the calculation of the PA_k_(t), noise amplification due to differentiation [54] has to be considered. Therefore, a Fourier analysis was conducted on the raw PP_k_(t), revealing that the highest power resided in frequencies around 2–4 Hz, but visible power was still found in the frequency range between 5 and 10 Hz. Before the differentiation, the PP_k_(t) were therefore filtered with a 3^rd^-order zero-phase 10 Hz-low-pass Butterworth filter. Furthermore, in order to ensure that the presented results were non-coincidental and not influenced by the filtering artifacts, all analysis procedures were repeated with several cutoff frequencies (in the range of 3–13 Hz) [11].

To determine bilateral asymmetries in the coordinative structure, i.e., in the composition of postural movements (hypothesis H1), we calculated the participant- and trial-specific *relative explained variance*, *rVAR*_k_, from the PP_k_(t) to represent how much (in percent) the postural variance in each PM_k_ contributed to total postural variance observed during a trial. The *rVAR*_k_ thus quantifies how important each PM_k_ is for the overall coordinative structure of the balancing movements [11,35,40]. 

To examine bilateral asymmetries in the control of movement components (hypothesis H2), we calculated two participant- and trial-specific variables characterizing the time-evolution of the PA_k_, the *number of zero**-crossings* (*N*_k_) and the *standard deviations of time between zero-crossings* (σ_k_) [11,32,35,40]. The *N*_k_ quantifies how often the PA_k_ change direction. One interpretation for a higher *N*_k_ could be that the postural control system intervenes more, i.e., a higher *N*_k_ suggests that the specific movement component was more tightly controlled. The σ_k_ was calculated and interpreted as a measure of the temporal regularity (lower σ) or irregularity (higher σ) of neuromuscular interventions in the control of the movement component.

Finally, to address validity considerations, the current study used a leave-one-out cross-validation to evaluate the vulnerability of the PM_k_ and the dependent variables to changes in the input data matrix [32]. The first eight PM_k_ proved to be robust and were therefore selected to test our hypotheses. 

### 2.5. Statistical Analysis 

All statistical calculations were conducted by using the software SPSS (IBM SPSS Statistics 24, SPSS Inc., Chicago, IL, USA). Shapiro–Wilk tests were performed to test for normality. Independent sample *t*-tests were used to assess sex differences in the demographic data. For normally distributed data, a mixed-factor ANOVA was used to test for effects of leg dominance (within-subject factor) or sex (between-subjects factor) and for leg dominance*sex interaction effects on the dependent variables (*rVAR_k_*, *N_k_*, and σ*_k_*) for the first 8 principal components (k = 1–8). The variable *rVAR* was in some higher-order components (*rVAR*_3–8_) not normally distributed. In these cases, the corresponding nonparametric tests (Wilcoxon for leg preference; Mann–Whitney for sex effects) were conducted. In order to control the family-wise error rate (8 comparisons for each dependent variable), the Holm–Bonferroni correction [55] was applied by adjusting the alpha level to α = 0.0063 in the first rank and 0.0072 in the second rank. The *p*-values smaller than *p* < 0.05 were interpreted as a statistical trend. We also report effect sizes η_p_² (ANOVA) and $r=z/\sqrt{N}$ (Wilcoxon) and observed power. 

## 3. Results

### 3.1. Characteristics of the First Eight Principal Movements (PM_1–8_)

All participants were able to complete the unipedal balancing trials on the multiaxial unstable platform without falling or stepping away. Table 2 lists the relative explained variances, *rVAR*_k_ (column 2), the main plane of motion (column 3), and a qualitative interpretation of the predominant movement patterns (column 4) of the first eight PMs, which together explained 94% of the overall postural variance. Figure 1 and Video S1 show visualizations of PM_1–8_.

### 3.2. Leg-Dominance Effects on the Composition of Single-Leg Balancing Movements 

For the composition of postural movements (Table 3), none of the *p*-values for effects of limb dominance in the *rVAR* variables sufficed (when applying the Holm–Bonferroni correction) to conclude that there were differences between the dominant and non-dominant limbs. However, balancing on the non-dominant leg showed the tendency of using more anteroposterior ankle sway (PM_2_), as compared to balancing on the dominant leg (*rVAR*_2_, *p* = 0.046). 

### 3.3. Leg-Dominance Effects on the Control of Movement Components 

Among the two variables characterizing the control of movement components (*N*_k_ and σ_k_), we found only one significant effect of leg dominance which passed the criteria of the Holm–Bonferroni correction (Table 4): substantially more changes in the direction of PM_3_ acceleration were found when balancing on the non-dominant leg, as compared to balancing on the dominant leg (*N*_3_: F_(1, 24)_ = 9.47, *p* = 0.005, η_p_^2^ = 0.283). However, when balancing on the non-dominant leg, we found a tendency (*p* < 0.05) toward a more tightly (higher *N*) and more regularly (lower σ) controlled movement in two movement components, PM_3_ and PM_6_. Strong effect sizes corroborate this finding. PM_3_ and PM_6_ both represent diagonal movement components, i.e., movement components that combined sagittal and frontal-plane movements. 

### 3.4. Sex Effects and Interactions between Sex and Leg Dominance 

No main effects of sex and no interaction effects between sex and leg preference were found. 

## 4. Discussion

The first hypothesis, H1, predicted that the effects of leg dominance might emerge as a difference in the coordinative structure of postural movements between the legs. The current findings do not allow an irrefutable conclusion on this hypothesis. No statistical significance was found; however, the observed power was small, which indicates a high probability for beta error [56]. Considering that we found a statistical trend and medium-to-large effect sizes [57], we suggest that our results may be useful as pilot data for future research. The second hypothesis, H2, predicted that leg dominance might emerge in different control characteristics in specific movement components. Our findings corroborate this hypothesis: We found a significant effect in PM_3_ and statistical trends for control variables derived from PM_3_ and PM_6_. That different control emerged in specific, but not in all, movement components agrees with the *minimal intervention principle*, as outlined in the introduction, and further supports our notion that analyzing movement components is more sensitive to altered control characteristics than traditional approaches. Finally, differences between the sexes or sex and leg-dominance interactions were not found.

Although our current results are inconclusive regarding leg-dominance effects on the coordinative structure of the balancing movements, there are a number of physiologic mechanisms that can be discussed when considering altered coordinative structure of movement. One mechanism might be an increased requirement of proprioceptive inputs. Previous work reported that high oscillations of a balance board might be associated with high gains of proprioceptive feedbacks [58], since the major kinesthetic sensor, muscle spindles, is particularly sensitive in detecting muscle length changes and muscle contraction velocities for sensing position and movement of the limbs [59,60]. Especially, when balancing is performed under difficult or threatening postural conditions, muscle spindle sensitivity is increased [61,62]. Thus, somatosensory information, which is perceived and integrated by the central nervous system (CNS) for controlling movements and stability, is increased [60,62]. Whether or not the neural systems controlling the dominant and non-dominant leg differ in their processing of sensory inputs, and whether this causes adaptations in the motor behavior, should be investigated in further studies.

The current results do suggest that the non-dominant controller does control the movement in PM_3_ more tightly (higher *N_3_*) than the neural control system of the dominant leg (lower *N_3_*). PM_3_ characterized an anterolateral hip strategy, and thus—and in contrast to PM_1_, PM_2_, PM_4_, PM_5_...—a diagonal-plane movement. Generally, diagonal-plane movements are more complex than single-plane movements, because they involve simultaneous movements at several joints and the activation of bi-articular muscles [63]. In addition, a previous study reported that, when performing diagonal-plane movement patterns, an increased beta-band energy (absolute power) was observed in the cortexes, suggesting a greater cortical organization in both hemispheres, as compared to single-plane movement patterns [64]. In the current study, the anterolateral trunk movement around the hip joint observed in PM_3_ was predominantly generated by the core musculature (spine, hips and pelvis, proximal lower limb, and abdominal structures) [65]. Control of core musculature is particularly relevant in providing proximal (core) stability when initiating distal mobility [65,66]. In this sense, if we assume that the non-dominant circuitry may be less suited for the more complex diagonal-plane movement, then this consideration would be in line with a tighter control, reflecting the increased difficulty in controlling core stability during single-leg balancing [67]. 

Regarding differences between the sexes, our findings were consistent with previous observations that movement structure and movement control characteristics do not differ between males and females in single-leg balancing on a level-rigid floor [11,35] or on a soft surface [35]. It seems that, in a physically active young population, although male participants were taller and heavier than females, the different body shapes and sizes did not lead to altered unipedal postural control.

Clinically, leg dominance has been discussed as a potential risk factor for sports-related lower-limb injuries [12,13,14,15]. Bilateral asymmetry in controlling diagonal-plane movement components involving the core movements should be considered for injury prevention and rehabilitation, especially for the non-dominant leg, since weakness in hip muscles and resulting alteration of hip/trunk position was associated with incidence of knee injury [65,68]. In this sense, core stability training on one leg is recommended and can be made a challenge, especially for athletes, by performing on an unstable surface.

### Limitations

The current study investigated differences in coordination and control between the dominant and non-dominant leg, defined as the preferred leg for kicking a ball [11,35]. Other definitions of leg dominance can be found in the literature [7,8,9,10,11], but our results are not necessarily applicable for other definitions. However, the kicking leg has often been used to classify the dominant leg, as reported in a recent review article [69], and proved to be more effective in determining interlimb differences in unipedal postural control than classifications of the dominant leg based on, for example, a static task [11]. 

Furthermore, the current study cannot attest for differences between left- and right-footed individuals, since this was not the purpose of the current study, and the number of left-footed participants in the current study is too small for such an analysis. However, the percentages of right- and left-footed participants in the current study is comparable to the percentages of right- and left-handedness reported for the world population [70,71]. Nevertheless, the number of volunteers of the current work (26 physically active young adults who reported no neuromuscular problems) was higher than the standard number of participants (10–22 participants) recruited in previous studies [8,16,17,18]. Furthermore, in the current study, we applied more rigorous statistical corrections for family-wise alpha-error accumulation. Thus, where we are able to report statistically significant differences, we can be confident in concluding on bilateral asymmetry in unipedal postural control.

A methodological limitation of the current study, which cannot be avoided, was that the control variables were computed from the second derivatives (PA_k_(t)) of principal position PP_k_(t) data. Time differentiation causes noise amplification and requires appropriate filtering of the data. We addressed this limitation by applying a Fourier analysis to select the optimal cutoff frequency for the current data and validated the results by performing repeated analyses with a variety of cutoff frequencies [11]. The current results consistently appeared with cutoff frequencies in the range of 8–13 Hz (Appendix A), supporting that they were non-coincidental. 

## 5. Conclusions

The current study corroborates the notion that leg dominance affects motor behavior when standing on an unstable balance board. Specifically, our results were inconclusive regarding differences in the coordinative structure of balancing movements; however, significant statistical differences were found in how diagonal-plane movement components are controlled. We propose that leg dominance should be considered when applying balance boards in physiotherapeutic or sports-related training settings and that particular attention should be given to movement dimensions outside the cardinal planes of motion.

## Figures and Tables

**Figure 1 brainsci-10-00128-f001:**
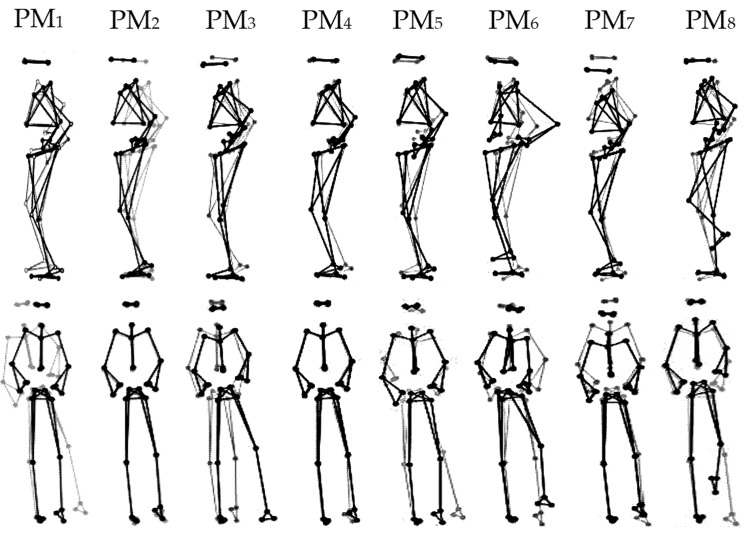
Visualization of the first eight principal movements (PM_1__–8_) analyzed from unipedal balancing on a multiaxial unstable platform. Note: Gray and black lines/dots show the extreme postures in opposite directions. For higher-order movement components, the movement amplitude was artificially amplified for a better visualization, using the amplifications 1× for PM_1__–2_, 2× for PM_3__–4_, 3× for PM_5–6_, and 4× for PM_7__–8_.

**Table 1 brainsci-10-00128-t001:** Characteristics of participants (mean ± SD; * *p* < 0.05).

	Male (*n* = 14)	Female (*n* = 12)	*p*-Value
Age (y)	25.8 ± 2.9	24.6 ± 5.3	0.478
Weight (kg)	77.5 ± 10.8	62.6 ± 4.9	<0.001 *
Height (cm)	180.0 ± 7.2	169.2 ± 4.3	<0.001 *
Body mass index (kg/m^2^)	23.9 ± 2.8	21.9 ± 2.3	0.059
Physical-activity participation (h/wk)	8.1 ± 5.5	8.8 ± 4.5	0.418

**Table 2 brainsci-10-00128-t002:** For the first eight principal movements (PM_1__–8_), the relative explained variance (*rVAR* (%)), the plane in which the PM motion mostly takes place, and a qualitative description of the PMs.

PM	*rVAR* [%]	Main Plane of Motion	Main Movements
1	44.2 ± 11.3	Frontal (Mediolateral)	Hip strategy coupled with ankle supination/pronation and knee flexion/extension of the stance leg, and hip adduction/abduction and flexion/extension of the lifted leg
2	26.0 ± 13.2	Sagittal (Anteroposterior)	Ankle strategy
3	7.6 ± 5.7	Diagonal (Anterolateral)	Hip strategy coupled with ankle supination/pronation of the stance leg, and the combination of hip abduction/adduction and knee flexion/extension of the lifted leg
4	7.0 ± 3.9	Frontal (Mediolateral)	Hip strategy coupled with knee flexion/extension and ankle supination/pronation of the stance leg, and knee flexion/extension of the lifted leg
5	3.2 ± 3.0	Frontal (Mediolateral)	Hip strategy coupled with ankle supination/pronation of the stance leg, and the combination of hip flexion/extension and abduction/adduction of the lifted leg
6	2.5 ± 1.6	Diagonal (Oblique- transverse)	Upper-body oblique rotation to the lifted leg coupled with hip flexion/extension of the lifted leg
7	2.0 ± 2.5	Vertical	Breathing movement patterns: upward/downward moving of the shoulders coupled with knee flexion/extension of the stance leg, and the combination of hip and knee flexion/extension of the lifted leg
8	1.6 ± 1.2	Vertical	Hip and knee flexion/extension of the lifted leg coupled with slightly upward/downward moving of the shoulder, and ankle dorsiflexion/plantarflexion of the stance leg

**Table 3 brainsci-10-00128-t003:** Comparing the relative explained variance (*rVAR* (%)) of the first eight principal movements (PM_1-8_) between the dominant (DO) and non-dominant (ND) legs (mean ± SD, ^a^ Repeated measures ANOVA, ^b^ Wilcoxon signed rank test; bold letters: *p*-values smaller than 0.05).

PM	*rVAR* (%)				
	ND	DO	*p*-Value	Effect Size	Observed Power
1 ^a^	41.1 ± 13.2	47.2 ± 11.5	0.061	η_p_² = 0.139	0.472
2 ^a^	29.3 ± 12.7	22.8 ± 13.5	**0.046**	**η_p_² = 0.156**	**0.524**
3 ^b^	7.8 ± 7.2	7.3 ± 5.8	0.970	r = 0.007	0.050
4 ^b^	6.9 ± 5.2	7.1 ± 3.9	0.949	r = 0.012	0.050
5 ^b^	3.1 ± 2.0	3.3 ± 3.1	0.949	r = 0.012	0.050
6 ^b^	2.2 ± 1.3	2.7 ± 1.6	0.269	r = 0.216	0.185
7 ^b^	1.7 ± 1.6	2.3 ± 2.6	0.151	r = 0.281	0.281
8 ^b^	1.5 ± 0.9	1.7 ± 1.2	0.568	r = 0.111	0.085

**Table 4 brainsci-10-00128-t004:** Comparing the number of zero-crossings (*N_k_*), and the standard deviation of the time between zero-crossings (σ*_k_* (milliseconds)), of the first eight principal accelerations (PA_1–8_) between the dominant (DO) and non-dominant (ND) legs (mean ± SD; * significance with Holm–Bonferroni correction, at *p* < 0.006; *p*-values smaller than 0.05 are printed bold).

**k**	***N*_k_ (Counts of PA_k_(t)-zero Crossings during 60 s)**				
	**ND**	**DO**	***p*-Value**	**Effect Size (η_p_^2^)**	**Observed Power**
1	441 ± 74	433 ± 67	0.642	0.009	0.074
2	527 ± 72	511 ± 74	0.204	0.066	0.241
3	516 ± 73	464 ± 75	**0.005 ***	**0.283**	**0.839**
4	531 ± 57	510 ± 68	0.144	0.087	0.305
5	531 ± 55	532 ± 56	0.832	0.002	0.055
6	565 ± 72	543 ± 72	**0.016**	**0.218**	**0.700**
7	509 ± 40	503 ± 45	0.444	0.025	0.116
8	519 ± 56	515 ± 53	0.632	0.010	0.075
**k**	**σ_k_ (milliseconds)**				
	**ND**	**DO**	***p*-Value**	**Effect Size (η_p_^2^)**	**Observed Power**
1	108.4 ± 28.9	113.6 ± 28.9	0.434	0.026	0.119
2	84.6 ± 18.4	86.7 ± 19.4	0.522	0.017	0.096
3	85.3 ± 18.3	96.3 ± 21.2	**0.035**	**0.172**	**0.573**
4	71.2 ± 15.6	74.9 ± 20.0	0.345	0.037	0.152
5	72.1 ± 13.6	72.4 ± 13.3	0.900	0.001	0.052
6	68.0 ± 14.2	74.6 ± 17.2	**0.025**	**0.192**	**0.630**
7	52.7 ± 8.4	53.6 ± 10.4	0.657	0.008	0.072
8	55.1 ± 7.5	56.1 ± 8.1	0.492	0.020	0.103

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
