# Peer review of "Leg Dominance Effects on Postural Control When Performing Challenging Balance Exercises"

_brainsci, 2020, doi:10.3390/brainsci10030128_

Round 1

Reviewer 1 Report

The authors present an interesting article detailing the use of principal component analysis (PCA) to identify whether differences in postural control exist between dominant and non-dominant legs during balance tasks. This work expands on the authors' previous work using PCA to quantify postural changes during other tasks. In the current work, optical motion capture was used to measure posture during one-legged balance tasks for 26 individuals, and the resulting posture data were broken into principal components to identify the "principal movements" (PMs) and "principal postures" characterizing the motion during the task. This process was developed in the authors' previously published research. While no significant differences were found in the overall contributions of the PMs between dominant and non-dominant stance, the results to suggest that two of the PMs exhibit more highly controlled movement. While the specific research question being asked is fairly limited and scope, the results do (as the authors state) provide compelling reason for additional research into

The article itself is well written, the methodology reasonable and thoroughly described, and the results and conclusions logically follow the data. I applaud the authors on a nicely prepared manuscript and an interesting technique for assessing postural control strategies.

My only minor comments are as follows:

Was any consideration given, or were participants screened for individuals exhibiting no leg preference or no leg dominance? Would it be beneficial to pre-screen or pre-evaluate the participants based on the dominance in one leg or another? It seems likely that the population exhibited a range of dominance of one leg over another that may have influenced the results. The article concludes that they found no main effects or interaction effects of leg preference on any of the dependent variables (Line 244). However, with 25/26 participants being right-leg dominant, the sample is highly skewed, and most likely too small to be able detect any effects of leg preference at all.  I understand this statement is not meant to be interpreted as a major finding of the study, but the wording should be careful to clarify that the lack of significance does not imply that no effects exist. The use of bold text in Table 4 is not clearly explained. Although the corresponding rows are called out in the text, it would be helpful to clearly state the significance of the bold format in the table caption.

Reviewer 2 Report

Title of the manuscript: “Leg Domin ance Effects on Postural Control when Performing Challenging Balance Exercises”

In this manuscript authors applied a principal component analysis (PCA) on kinematic data to investigate, from a side, the bilateral asymmetry on the coordinative structure (hypothesis H1), and, from the other side, the bilateral asymmetry on the control characteristics of specific movement components (hypothesis H2). Experiments of marker-based motion tracking was performed on 26 healthy adults, who stood unipedally on a multiaxial-unstable board in a randomized order on their dominant and non-dominant leg. Results were inconclusive regarding a leg dominance effect on the coordinative structure of balancing movements (H1 inconclusive). Nevertheless, investigating principal movements due to a diagonal plane movement component, H2 was supported (p = 0.005). Authors claim that these findings supported that leg dominance effects should be considered when assessing or training lower-limb neuromuscular control, ggesting that specific attention should be given to diagonal plane movements

General comment: “This manuscript seems to be written adequately. The logic flow of the text is quite linear. Nevertheless, some points of the manuscript, as well as the overall value of this work, are not clear. In particular, since the current study applied a principal component analysis (PCA) to kinematic data to extract PPs and PAs, the physical meaning of this procedure should be carefully explained together with all relevant hypotheses and limitations. Indeed it is not clear whether the proposed PCA procedure is able to extract relevant physical information or each PC is a simple combination of kinematic parameters, without a clear physical meaning. In addition, the value of this work with respect to the current literature should be better underlined by authors, as well as its innovative implications with respect to relevant fields as injury prevention and rehabilitation.

Some specific comments:

Lines. “Footedness is also interesting from an epidemiological point of view since leg dominance has been reported as an internal risk factor for sports-related lower-limb injury [12–15] “

*) This is interesting, perhaps authors could expand this issue within the discussion section.

Lines: “Only in a body-tracking test [19], effects of footedness were observed when participants received visual feedback of their performance”

*) Authors could explain why the visual feedback is important.

Lines: “Since training on unstable support surfaces is an important therapeutic exercise with proven effectiveness in facilitating and regaining neuromuscular control [34–38], investigating unipedal balancing movements on a multiaxial unstable platform is of particular interest and may provide clinically relevant information for injury prevention and rehabilitation.

*) These lines are interesting, perhaps authors could expand this issue within the “Discussion” section.

Could the results of this work be applied to injury prevention and rehabilitation ? Please explain within the “Discussion” section.

Lines:” Leg dominance was defined as the preferred leg for kicking a ball since this definition had proven most effective in determining interlimb differences in unipedal postural control [11,33]. Most participants (96%) self-reported right leg dominance. The leg dominance of all participants coincided with their hand dominance”.

*) As reported in the previous text: “Most participants (96%) self-reported right leg dominance”. Nevertheless, authors claim that: “Results were inconclusive regarding a leg dominance effect on the coordinative structure of balancing movements (H1 inconclusive)”. Therefore, authors should explain this apparent discrepancy. What are the main motivation of this ? There are weaknesses within the proposed framework ? All these points should be clearly developed within the “Discussion” section.

Lines: “One PCA-analysis was conducted after normalizing, weighing, and pooling the middle 60 seconds of each balancing trial. Specifically, the following pre-processing steps were conducted: nine asymmetrical markers placed on upper arms, lower arms, right scapular, upper thighs, and the lower thighs were omitted [11,33].”

*) Authors could better explain why the middle 60 seconds of each balancing trial were used to perform the analysis.

Lines:” All left-leg trials were mirrored and relabeled such that all input appeared as right-leg trials;
thus, only one PCA could be computed and the results could be directly compared between the
dominant and non-dominant legs [11,33]. Then, all marker data were centered by subtracting the
mean posture vector [41–43], thus preventing differences in mean marker positioning in space
having an influence on the PCA-outcome [20]. The centered posture vectors were normalized using
the mean Euclidean distance [20,22] to address anthropometric differences [20], and weighted by
considering sex-specific mass distributions for yielding comparison between the sexes [44,45].”

*) Authors could better explain these lines. In particular, it is not totally clear why “left-leg trials were mirrored and relabeled such that all input appeared as right-leg trials; thus, only one PCA could be computed and the results could be directly compared between the dominant and non-dominant legs [11,33]”. In addition, it is not clear how markers were fixed and what was the magnitude of errors about their positions during the recorded trials. Finally, authors could add some specific information about the following numerical procedures: ”The centered posture vectors were normalized using the mean Euclidean distance [20,22] to address anthropometric differences [20], and weighted by considering sex-specific mass distributions for yielding comparison between the sexes [44,45].”

Lines: “ The PCA was computed through a singular value decomposition of the input matrix’s covariance matrix [41–43]. The PCA yielded a set of eigenvectors, which form a new basis for the
vector space of marker positions, i.e., the eigenvectors provide a basis in which changes in posture
(“postural movements”) can be quantified [41–43]. Since the PCA eigenvectors represent correlated
changes in the underlying data, the eigenvectors can be interpreted as patterns of correlated segment
movements, as “movement strategies” [20,22] or “principal movements” (PMs) [23]. Through
projecting the normalized posture vectors recorded for each trial onto the eigenvector basis, we
obtained the PCA scores, which we call “principal positions” PPk(t), where t denotes the time and k
denotes the order of the PM. The PPk(t) quantify how much the posture at time t deviates from the
mean posture according to the pattern defined by the k-th eigenvector. Thus, the PPk(t) represent the
amplitude of each movement component PMk. The term “principal” in the variable name indicates
that the variable was derived from the PCA, i.e., that the variable is defined through the PCA
eigenvector basis.

*) Authors should better explain the physical meaning of the “principal movements” (e.g., in terms of markers position over time or other meaningful physical variables) and the usefulness (from a physical point of view) of the projection of the normalized posture vectors recorded for each trial onto the eigenvector basis.

Lines: “In analogy with Newton’s mechanics, the PPk(t) can be differentiated to obtain “principal
velocities” and “principal accelerations”, PAk(t) [23]. The PAk(t) quantify the acceleration of the
postural movements and thus represent a variable that quantifies how the mechanical system is
controlled [23]. In unperturbed balancing trials, there are only two types of forces that produce
postural accelerations: gravity (which is constant) and forces produced by the sensorimotor system.
In other words, together the PAk(t) represent the resultant control variables through which the
postural control system changes its kinematic state [38].

*) Authors should better explain the “analogy with Newton’s mechanics”. Indeed, the differentiation with respect to the time could be meaningful if (and only if) the variable has a physical meaning. Perhaps the differentiation of a linear combination of displacements could not have a clear physical meaning...

Lines: “To determine bilateral asymmetries in the coordinative structure, i.e., in the composition of
postural movements (hypothesis H1), we calculated the participant- and trial-specific relative
explained variance rVARk from the PPk(t) to represent how much (in percent) the postural variance in each PMk contributed to total postural variance observed during a trial. The rVARk thus quantify,
how important each PMk is for the overall coordinative structure of the balancing movements
[11,33,38]. “

*) Since a crucial conclusion of the manuscript is based on PPi(t), see the previous concern about the physical meaning of PPi(t). Could authors show that this procedure is physically consistent ?

Could the consistency of this framework be related to the apparent discrepancy between initial data “ Most participants (96%) self-reported right leg dominance” and results “Results were inconclusive regarding a leg dominance effect on the coordinative structure of balancing movements (H1 inconclusive)” ? Please, explain within the “Discussion” section.

Lines: “To examine bilateral asymmetries in the control of movement components (hypothesis H2), we calculated two participant- and trial-specific variables characterizing the time-evolution of the PAk, the number of zero-crossings (Nk) and the standard deviations of time between zero-crossings (σk) [11,30,33,38]. “

*) See the previous comment and the concerns about the physical consistency of the differentiation with respect to the time of PPi(t). Please, explain within the “Discussion” section.

Round 2

Reviewer 2 Report

Title of the manuscript: “Leg Domin ance Effects on Postural Control when Performing Challenging Balance Exercises”

In this manuscript authors applied a principal component analysis (PCA) on kinematic data to investigate, from a side, the bilateral asymmetry on the coordinative structure (hypothesis H1), and, from the other side, the bilateral asymmetry on the control characteristics of specific movement components (hypothesis H2). Experiments of marker-based motion tracking was performed on 26 healthy adults, who stood unipedally on a multiaxial-unstable board in a randomized order on their dominant and non-dominant leg. Results were inconclusive regarding a leg dominance effect on the coordinative structure of balancing movements (H1 inconclusive). Nevertheless, investigating principal movements due to a diagonal plane movement component, H2 was supported (p = 0.005). Authors claim that these findings supported that leg dominance effects should be considered when assessing or training lower-limb neuromuscular control, ggesting that specific attention should be given to diagonal plane movements

Comments:

Authors revised the manuscript: this is laudable. However, within the "Limitations" paragraph perhaps authors could say "However, the percentage of right- and left- footed participants in the current study is comparable to the percentages of right- and left-handedness people reported for the world population [69,70]."

Nevertheless, the statistical significance of this work is due to the amount of volunteers: "Twenty-six physically active young adults who reported no neuromuscular problems".

A brief discussion about the statistical significance of this work (with respect to the standard number of people recruited in other studies), could be beneficial to convince the reader about the importance of results.

In addition, if "The current study investigated differences in coordination and control between the dominant and non-dominant leg, defined as the preferred lag for kicking a ball. Other definitions of leg dominance can be found in the literature, but our results are not necessarily applicable for other definitions." interested readers could understand that all the claimed results depends on the definition of "leg dominance".

Therefore, authors should discuss more in depth the general validity of their claimed results.
